# Spatial Raman Spectroscopy to Characterize (Sulfated) Glycosaminoglycans in Human Articular Cartilage

**DOI:** 10.3390/ijms26209875

**Published:** 2025-10-10

**Authors:** Andrea Schwab, Jannik Jahn, Kerstin Sitte, Christoph H. Lohmann, Jessica Bertrand, Sonja Gamsjaeger

**Affiliations:** 1Department of Orthopaedics, Medical Faculty, Otto-von-Guericke University, 29120 Magdeburg, Germany; 2Institute for Forensic Medicine, University Hospital Halle, 06120 Halle (Saale), Germany; 3Ludwig Boltzmann Institute of Osteology, Hanusch Hospital of OEGK and AUVA Trauma Centre Meidling, 1st Medical Department, Hanusch Hospital, 1140 Vienna, Austria

**Keywords:** articular cartilage, osteoarthritis, Raman spectroscopy, skeletal development, sulfation, glycosaminoglycans

## Abstract

Raman spectroscopy allows for analyzing local molecular matrix components while preserving spatial resolution in tissue samples. The aim of this study was to use Raman line scans to discriminate between healthy and diseased cartilage tissue based on the depth-dependent sulfated glycosaminoglycans (sGAG) and total GAGs distribution. Full-thickness articular cartilage tissue was harvested from human individuals at different maturation stages (skeletally immature, skeletally mature) and from patients with diagnosed osteoarthritis. Raman spectroscopic line scans (30 µm step size) were utilized to analyze the sub-zonal sGAG (1062 cm^−1^) and total GAG (1370–1380 cm^−1^) distribution relative to the organic matrix (CH_2_ band 1430–1480 cm^−1^). We found a linear trend of the sGAG/CH_2_ ratio over the tissue depth in all samples (*p* < 0.0001). The total GAG/CH_2_ ratio of the skeletally immature and mature cartilage showed a characteristic non-linear behavior over the tissue distance. The elderly osteoarthritic cartilage exhibited lower total GAG/CH_2_ ratios compared to the ratios of the skeletally immature and mature samples, without a pronounced increase in the superficial area. Raman spectroscopic line scans are a fast and representative method allowing us to identify the local and tissue depth-dependent distribution of GAGs at higher specificity and resolution compared to histological staining.

## 1. Introduction

Articular cartilage, the tissue covering the joints, is a structurally complex tissue, organized hierarchically at both the matrix and cellular scales. It is composed of the superficial, the middle, and the deep zones. These zones vary in the composition of the extracellular matrix (ECM) and the orientation of the collagen fibers [1,2,3]. The zonality of the articular cartilage is formed at skeletal maturity [4]. The main components of the ECM in immature and mature articular cartilage are collagen type II and proteoglycans [5,6]. Collagen type II is a fibrillar collagen that forms the characteristic arch-like structure referred to as Benninghoff arcades. The collagen type II content in articular cartilage decreases from the articular surface (superficial zone) towards the middle zone and the deep zone and anchors through the calcified cartilage layer in the subchondral bone [1]. Proteoglycans are the main non-collagenous component of the ECM of articular cartilage. Proteoglycan macromolecules consist of a core protein with one or more covalently attached glycosaminoglycan (GAG) chains. GAGs are naturally modified by sulfation groups or carboxy groups, resulting in the negative charge of these macromolecules. Chondroitin sulfate is the main sulfated GAG (sGAG) in aggrecan, a proteoglycan mainly found in the interterritorial ECM of articular cartilage. Hyaluronan is the only non-sulfated GAG, thus the only neutral GAG in articular cartilage, and functions as a backbone where proteoglycans are non-covalently bound to [6,7].

During cartilage development and in cartilage disease, changes in the composition of collagen fibers and their orientation and collagen type, as well as proteoglycan content and the sulfation of sGAGs, have been described [8]. Specifically, the chondroitin sulfate sulfation pattern varies in human articular cartilage across different zones with skeletal maturity and age. During skeletal development, disulfated residues are mainly located at the C-4 and C-6 positions of sGAGs, while the C-6 sulfation is more prominent in sGAGs in skeletally mature adult cartilage [9,10].

One of the most prevalent cartilage diseases in elderly people is osteoarthritis (OA), a degenerative joint disease. Pathological changes associated with OA include the degradation of sGAGs and cartilage fibrillation, resulting in tissue degeneration. The degradation of sGAGs in the superficial zone is one of the earliest pathological hallmarks in OA. The formation of fibrous collagen and the progression of sGAGs degeneration towards the middle and deep zones of articular cartilage are characteristic of OA disease progression [11,12]. These changes have been visualized and described based on histological stainings and biochemical assays (e.g., hydroxyproline assay, sGAG assay) or high-performance liquid chromatography [13]. While histological stainings are not sensitive enough to quantify intra-zonal changes, biochemical assays and liquid chromatography are sensitive but lack the spatial resolution. Therefore, spatially resolved analyses, including Raman spectroscopy or Fourier-transform infrared spectroscopy, are promising tools to characterize and identify local changes in ECM components in health and disease in zonally structured tissues [14,15,16,17,18,19,20,21]. Raman spectroscopy enables localized molecular analysis with high chemical specificity while preserving the spatial resolution of the tissue architecture and allows for detailed analysis of biochemical components within intact tissue samples, making it a valuable tool for investigating structural and compositional changes in engineered or native cartilage. Raman spectroscopy relies on the inelastic scattering of photons, where incident light interacts with molecular vibrations, leading to energy shifts that reveal the vibrational modes of chemical bonds.

We and others have demonstrated that Raman spectroscopic imaging enables the detection of GAGs in cartilage and the identification of mineral components, such as phosphate and carbonate, in bone tissue [16,22,23,24]. Raman spectroscopic imaging has been used to detect sGAGs from the peak at 1062 cm^−1^ (symmetric stretching vibration of the OSO_3_ cm^−1^ group), total GAGs from the band at 1370–1380 cm^−1^ (unbranched chains of repeating sugar molecules, polysaccharides, and the CH_3_ symmetric deformation), and organic matrix components (CH_2_ band: 1430–1480 cm^−1^) for the distinction of tissue types and to characterize GAG loss in articular cartilage in OA patients [15,22,25,26,27]. Raman spectroscopic results are highly informative spatially resolved data showing the local distribution of the ECM components. The main disadvantage of Raman spectroscopic imaging is the inherently time-consuming procedure, due to the need for point-by-point spectral acquisition across a sample to achieve high resolution. An alternative to Raman imaging is Raman spectroscopic line scans. Raman line scans enable spatially resolved data collection along defined paths and with high resolution across the tissue, thereby capturing structural and compositional heterogeneity more efficiently while significantly reducing acquisition time compared to Raman imaging. We hypothesize that Raman spectroscopic line scans allow the identification of tissue depth-dependent local ECM changes within the different zones of articular cartilage, enabling discrimination between immature, mature, and diseased articular cartilage tissue.

In this study, we used spatially resolved Raman spectroscopic line scans to measure the zonal and sub-zonal distribution of sGAGs and total GAGs (sulfated and non-sulfated) in the superficial layer, middle zone, and deep zone of human articular cartilage tissue. We propose that the ratio of sGAGs/CH_2_ and the total GAG/CH_2_ are indicators to discriminate between healthy and diseased cartilage and to estimate the maturity of articular cartilage.

## 2. Results

### 2.1. The Total GAG/CH_2_ Ratio over the Tissue Distance Is an Indicator to Estimate the Maturity of Human Articular Cartilage

We used Raman spectroscopy to calculate the ratio of total GAGs and sGAGs relative to the organic matrix (CH_2_ band) in intact and degenerated articular cartilage at a step size of 30 µm and a resolution of 1 × 1 µm (measurement area). The distance was calculated as a percent of the total thickness of each tissue, with 0% referring to the cartilage surface (superficial layer) and 100% to the cartilage deep zone (deep zone cut off from the subchondral bone). We investigated the sub-zonal total GAG and sGAG distribution in articular cartilage at different stages of skeletal maturity (Figure 1). To evaluate the reproducibility of the sGAG/CH_2_ and total GAG/CH_2_ ratios, three line scans per sample were performed, showing the biological variability within one sample with an overall low variability and good reproducibility. The sGAG/CH_2_ ratios of the cartilage from immature, mature, and elderly donors showed a linear increase and positive correlation across the tissue depth (*p* < 0.0001). The two cartilage samples from the young individuals (4 years and 21 years) exhibited a similar linear increase, while the sGAG/CH_2_ ratio of the mature and elderly patient (77 years) resulted in higher sGAG/CH_2_ ratios compared to the tissue samples from the young individuals at all distances (Figure 1A). A non-linear correlation (one-phase decay) over distance, reaching a plateau at 50% of the total distance, was characteristic of the total GAG/CH_2_ ratio in the skeletally immature (4 years) and young mature (21 years) cartilage tissue. The elderly mature sample (77 years, macroscopic intact cartilage) showed a less pronounced increase in total GAG/CH_2_ over the tissue depth compared to the young cartilage tissue samples (Figure 1A).

The representative spectra (normalized to the respective CH_2_ band) of the superficial layer, middle zone, and deep zone of the three cartilage samples visualize the change in peak height for sGAGs and total GAGs (Figure 1B–D). The spectra of the superficial layer of the immature and young mature cartilage showed the least pronounced sGAG peak compared to the respective bands in the middle zone and deep zone. The band of the total GAGs was most pronounced in the middle zone of the immature and young mature tissue samples. The spectra obtained from the immature and mature young adult samples were smoother compared to the spectra of the mature elderly cartilage sample, a possible indicator for tissue degeneration. In the cartilage sample of the 77-year-old patient, the peak intensity of the sGAG band was higher compared to the sGAG peak in the younger cartilage samples (4 years and 21 years). This data highlights the potential of Raman spectroscopy to detect local changes in sGAGs and total GAGs within each of the three zones in articular cartilage.

**Figure 1 ijms-26-09875-f001:**
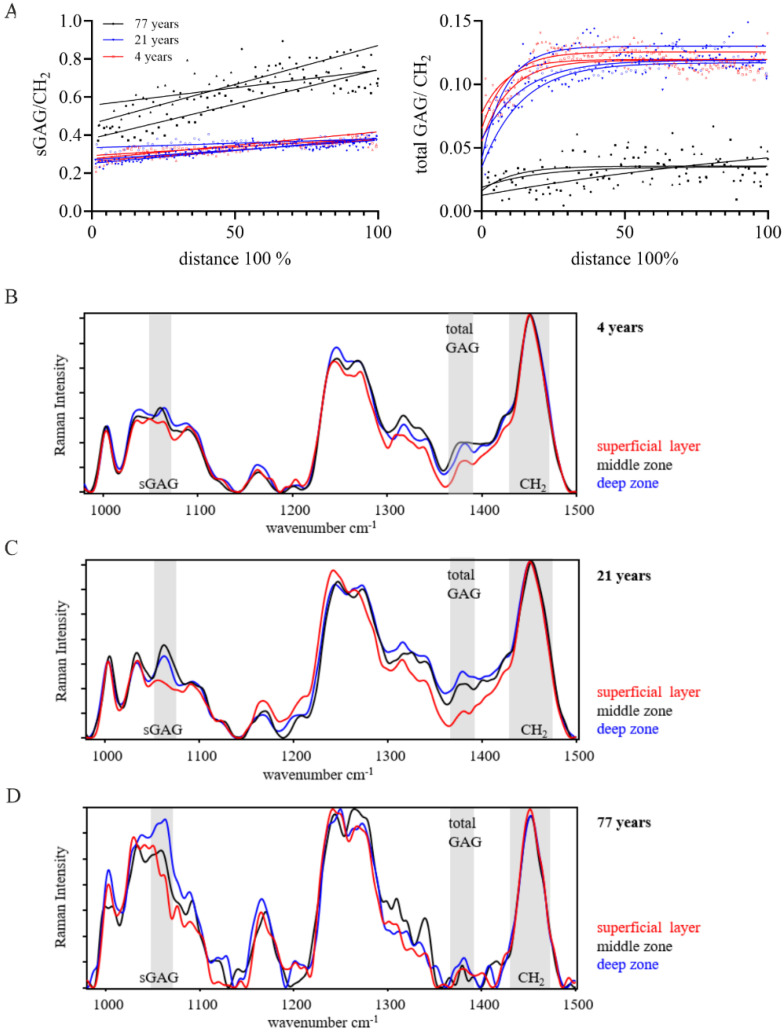
Raman spectra in human articular cartilage during development (skeletally immature, 4 years, red), young adult (skeletally mature, 21 years, blue), and mature elderly (77 years, black). (**A**) The ratio of the sGAG to the CH_2_ band showed a linear correlation with the cartilage depth in all samples (for 4 years, 21 years, and 77 years). sGAG/CH_2_ ratios in skeletally mature and immature samples were at comparable levels and increased from superficial towards the deep zone, while the ratio of the adult diseased samples showed higher ratios at all distances. The slope of the linear regression was different between samples (*p* < 0.0001). The total GAG/CH_2_ ratio in immature and mature cartilage tissue was approximated by a non-linear fit (on phase decay) with an increase over the tissue depth and reached a plateau at 50% distance. The total GAG/CH_2_ values of the cartilage tissue of the 77-year-old patient were fluctuating around the plateau without the characteristic steep increase in the superficial layer. Raman spectra normalized to the respective CH_2_ band of (**B**) immature articular cartilage, (**C**) mature non-OA cartilage, and (**D**) diseased cartilage are representative spectra of the superficial layer (red), middle zone (black), deep zone (blue). sGAG, total GAG, and the CH_2_ band are highlighted in gray. Number of measurement points for line scans: 4 years (72/71/78), 21 years (83/76/61), 77 years (42/47/39).

The histological staining of the skeletally immature, mature young, and mature elderly cartilage (Figure 2) confirmed the trend of the sGAG/CH_2_ ratio based on Raman spectroscopy shown in Figure 1A. However, the visual changes and sub-zonal sGAG distribution are not visible on histological staining due to the lower resolution compared to Raman line scans.

Articular cartilage from an immature (4 years) and mature (young adult, 21 years) patient showed a homogenous staining of the sGAGs in the Safranin-O and Alcian PAS staining (Figure 2A,B) with no differences between the two stainings. Strong sulfated sGAGs (blue staining in Alcian PAS staining) were located in all three zones of the tissue.

The histological stainings of the cartilage harvested from a mature and elderly patient (77 years) showed mild to moderate signs of OA on the Safranin-O staining (Figure 2C). The OA characteristic of sGAG leaching from the superficial layer down to the middle zone was present on the Safranin-O staining, while strong sulfated sGAGs were partially retained in the middle zone in the Alcian PAS staining. The H&E staining did not show differences in tissue morphology.

### 2.2. The sGAG/CH_2_ Ratio over the Tissue Distance Is an Indicator to Estimate the Health of Human Articular Cartilage

The sGAG/CH_2_ ratio showed a depth-dependent linear correlation in both OA samples (59-year-old patient), the macroscopically intact and degenerated articular cartilage, with a linear increase in the sGAG/CH_2_ ratio in the degenerated cartilage tissue (*p* < 0.0001, Figure 3A). In contrast, the macroscopic intact cartilage showed a decrease in this ratio (*p* < 0.0001) with increasing tissue depth (Figure 3A). Due to the breakdown of protein crosslinks in proteoglycans in the pronase-treated macroscopic intact cartilage, the sGAG/CH_2_ ratio was lower compared to the donor-matched non-treated sample, with no change in sGAG/CH_2_ over the tissue depth (*p* < 0.0001). Interestingly, the total GAG/CH_2_ ratio of all samples fluctuated below the GAG/CH_2_ ratio of 0.05, without the steep increase characteristic of immature and young mature cartilage tissue (Figure 3A).

The representative spectra of the superficial layer, middle zone, and deep zone of the three cartilage samples were normalized to the respective CH_2_ band (Figure 3B–D). The sGAG peak height increased from the superficial layer towards the deep zone in the intact and degenerated cartilage. In the degenerated cartilage sample, the intensity of the sGAG peak decreased from the deep zone towards the superficial layer. The total GAG peak intensity of the intact and degenerated cartilage remained at a lower level compared to the sGAG peak, with no total GAG peak in the superficial layer of the degenerated sample. The pronase treatment showed a reduction in the sGAGs and total GAGs peaks compared to the intact sample, thus confirming the removal of GAGs using the enzymatic treatment with pronase.

Histological stainings were used to visualize the distribution of sGAGs and tissue morphology in the articular cartilage of location- and donor-matched articular cartilage tissue samples used for Raman spectroscopy. The Safranin-O and Alcian PAS staining showed a loss of sGAGs in the superficial layer of macroscopically intact cartilage (Figure 4A), a sign of mild OA. Non-sulfated GAGs were located in the superficial layer in the intact sample (purple staining in the Alcian PAS staining, Figure 4A). First signs of superficial fibrillation and GAG loss were visible in the histological stainings.

**Figure 3 ijms-26-09875-f003:**
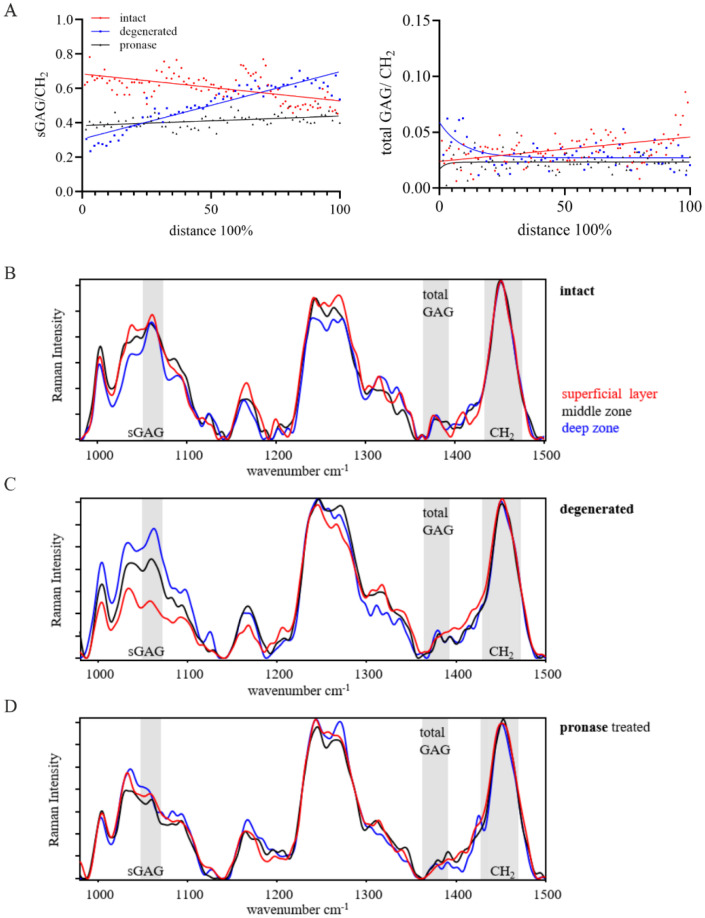
Raman spectra in human articular cartilage of intact cartilage (red), OA degenerated cartilage (blue), and pronase-treated cartilage (black). (**A**) The ratio of the sGAG/CH_2_ band showed a linear correlation with the cartilage depth in all three samples. The slope of the linear regression was different between samples (*p* < 0.0001). OA cartilage showed an increase and the intact cartilage a decrease in sGAG/CH_2_ over the cartilage distance. No change in sGAG/CH_2_ resulted for the pronase-treated tissue. The ratio of the total GAGs/CH_2_ band did not change with tissue depth. Raman spectra of (**B**) intact cartilage, (**C**) OA degenerated cartilage, and (**D**) pronase-treated intact cartilage, each displaying representative spectra of the superficial layer, middle zone (blue), deep zone (black). Peak height and respective area are increased from the superficial layer towards the deep zone of the intact and degenerated OA cartilage. Representative Raman spectra are normalized to the CH_2_ band, sGAG, total GAG, and CH_2_ band are highlighted in gray. Number of measurement points for line scans: intact (104), degenerated (64), and pronase-treated (71).

The macroscopically degenerated cartilage exhibited the characteristic fibrillation of the cartilage’s top layer in the fissures reaching into the middle zone (Figure 4B), indicating a progressed OA phenotype. The sGAGs were depleted in the area of the fissures, as noted in the Safranin-O staining.

**Figure 4 ijms-26-09875-f004:**
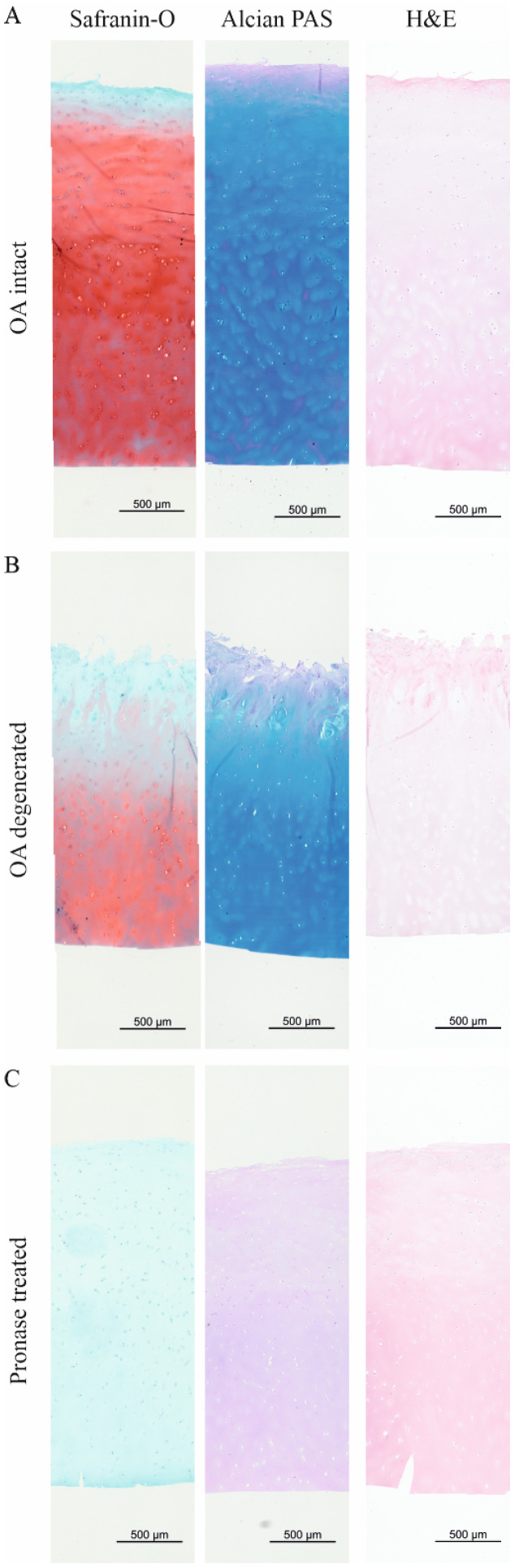
Histological stainings to visualize sGAGs and neutral GAGs in an (**A**) macroscopically intact, (**B**) pronase-treated intact, and (**C**) macroscopically degenerated OA cartilage. Safranin-O staining visualizes all sGAGs in red. Alcian PAS staining (pH 1.0) visualizes and differentiates strong sulfated sGAGs (blue) and neutral GAGs (purple). In intact cartilage, sGAGs showed a homogenous distribution throughout the cross-section, except for the sGAG-depleted superficial layer in both stainings. Pronase treatment successfully removed sGAGs. Degenerated cartilage showed sGAGs only in the deeper layers. In degenerated OA cartilage, Safranin-O staining showed sGAG degeneration down to the middle zone, while strong sulfated sGAGs (Alcian PAS staining) remained in the middle zone. Neutral stained GAGs (purple) are mostly present in the ECM in the fibrillated superficial layer. Tissue morphology (H&E staining) did not show obvious differences between the groups. Scale bar: 500 µm.

The strong negatively charged sGAGs visualized in the Alcian PAS staining remained in the middle zone, reaching toward the proliferation clusters in the fibrillated area in the superficial layer. Neutral GAGs were mainly found in the fibrillated tissue of the fibrillated superficial layer (Figure 4B). In contrast to spatially resolved Raman spectroscopy, histological stainings did not allow visualization of the zonal differences in sGAGs or total GAGs, thus emphasizing the higher sensitivity of Raman spectroscopy compared to routine stainings.

The Safranin-O and Alcian PAS staining confirmed the removal of sGAGs after the pronase treatment of intact cartilage (Figure 4C). The Alcian PAS staining showed a homogenous purple staining throughout the section, corresponding to non-sulfated GAGs, including hyaluronan, that remained in the cartilage ECM after enzymatic digestion.

## 3. Discussion

In this study, we introduced spatial Raman spectroscopic line scans to locally quantify sGAG (1062 cm^−1^) and total GAGs (1370–1380 cm^−1^) in articular cartilage tissue at different skeletal maturity and disease severity. Our data suggest age- and disease-specific depth-dependent differences in the sGAG and total GAGs relative to the organic matrix (CH_2_ band, 1430–1480 cm^−1^). To balance the demand for spatial resolution with practical constraints on measurement duration, Raman spectroscopic line scans were adopted in this study rather than full-area mapping or random single-point measurements. By scanning along lines, a broader coverage and a more representative assessment of the sample’s biochemical variability is generated, while significantly reducing the acquisition time compared to Raman imaging.

The identification of sGAG and total GAGs, together with the sub-zonal spatial resolution (30 µm step size and 1 × 1 µm measurement point in this study), highlights the sensitivity of Raman spectroscopy in characterizing the major non-collagenous components in articular cartilage. This higher sensitivity allows to detect sGAG loss in cartilage samples with small focal lesions before macroscopic changes are visible [24]. However, previous studies did not quantitatively analyze the ratios of sGAG and total GAG relative to the organic matrix or another ECM component to correct for background signals due to autofluorescence, allowing for comparison of the distribution in different samples [15,22,25,26,27]. In line with the literature, we demonstrated that enzymatic digestion removed GAGs from the cartilage tissue, thereby highlighting the specificity of the sGAG and GAG bands in the Raman spectra [15]. Previous studies by Bansil et al. and Ellis et al. have demonstrated the feasibility of distinguishing between various proteoglycans and GAGs using Raman spectroscopy with well-defined standard references [25,26]. These biomolecules exhibit distinct Raman spectral features, including band shifts and the presence of additional Raman peaks that enable their differentiation in simplified systems. However, applying these spectral signatures to complex biological tissues such as articular cartilage remains a substantial challenge. Dominant signals from the organic matrix [19,28], particularly those of collagen, obscure the characteristic bands of these biomolecules, making direct identification impossible. Therefore, Raman spectroscopic analysis does not allow for the differentiation between the various sGAG side chains (e.g., chondroitin sulfate, keratan sulfate, heparan sulfate) in tissues such as cartilage [15]. Instead, all sGAGs and total GAGs, which are represented by the sGAG peak at ~1062 cm^−1^ and the GAG content from the Raman band at 1370–1380 cm^−1^, can be separated in the Raman spectra obtained from cartilage. Despite enzymatic treatment with pronase, a Raman band for sGAGs and total GAGs remains detectable in the spectra. The low Raman signal suggests that the tissue contains residues of sGAGs not being detectable on histological sections. This observation highlights the high sensitivity of Raman spectroscopy over histological staining by detecting residual molecular features after biochemical tissue modification.

Raman spectroscopy, using line scans and Raman spectroscopic imaging, provides additional insight into the distribution of GAG and sGAG throughout the various zones of articular cartilage, whereas standard biochemical assays and histological stainings are limited to quantifying and visualizing sGAGs. Gao et al. used a similar range of the sGAG and GAG bands on OA cartilage to map the distribution of the two components as an indicator of cartilage degradation in a Raman imaging setup [16]. This approach was also used to map the zonal organization in tissue-engineered cartilage samples, highlighting the versatility of spatial Raman spectroscopy for biological and medical applications [14]. Raman imaging, in contrast to Raman line scanning, requires a point-by-point spectral acquisition across the sample at the desired resolution, which is inherently time-consuming. Scanning times of one day or more are often required to scan and map larger tissues using a Raman-imaging setup, including articular cartilage cross-sections. A total of 8–10 Raman images for a 2.5 mm thick cartilage sample would be necessary to cover all cartilage zones using a 20× objective. For obtaining high-quality Raman spectra and accurate quantification of biomolecules, it is essential that the signal-to-noise ratio be low, especially for small Raman bands. Employing an acquisition protocol consisting of a 5 sec exposure time per measurement point, repeated over 10 iterations for the Raman line scans. This setup would tremendously increase the time for Raman imaging.

We propose analyzing the total GAG/CH_2_ ratio and sGAG/CH_2_ ratios as indicators of cartilage skeletal maturity and health. The depth-dependent linear behavior of sGAG/CH_2_ represents the zone-dependent sGAG content in articular cartilage with an increase in sGAGs from the superficial zone towards the deep zone in healthy cartilage, while the decrease in this ratio is an indicator for cartilage degeneration with age and disease [29,30]. Interestingly, the total GAG/CH_2_ ratio, which quantifies the relative amount of sGAGs and non-sulfated GAGs, but not the sGAG/CH_2_ ratio, exhibited a tissue maturity-dependent non-linear behavior with an exponential increase in total GAGs in the superficial layer and middle zone. This non-linear behavior was characteristic of skeletally immature and skeletally mature cartilage in a young adult and was absent or less pronounced in elderly and OA-diseased articular cartilage samples. The age- and OA-related remodeling of the cartilage ECM, along with tissue degradation starting from the superficial layer, explains this behavior [11,12]. These observations are consistent with our findings, reinforcing the trend of increased sGAG level and reduced GAG content with age and disease. Histological stainings are not as specific in discriminating between sGAGs and non-sulfated GAGs, nor do they allow for the analysis of sub-zonal differences in the various zones of articular cartilage, highlighting the advantage of Raman spectroscopic line scans in identifying local and sub-zonal molecular fingerprints.

One limitation of Raman spectroscopy is that standard spectrometers (e.g., Senterra I, Bruker, used in this study) are typically equipped with a microscope, require sample preparation, and are not suitable for direct analysis of live tissue in vivo. Extended acquisition times required for Raman line scans (and even more so for Raman imaging) pose a challenge for measuring fresh biological tissue. During the procedure, the tissue undergoes drying and shrinkage, leading to morphological changes that can compromise the accuracy of spectroscopic data. To mitigate this, we preconditioned the samples by allowing them to dry overnight under carefully controlled environmental parameters. This protocol stabilizes the tissue morphology and enhances the reproducibility of the Raman measurements. To address these challenges, Bergholt et al. developed a diffuse fiber-optic Raman spectroscopy system that enables non-destructive, online monitoring of live cell-engineered cartilage growth [31]. This setup allows for the in situ quantification of ECM components, including collagen and GAGs. By eliminating the need for sample preparation and enabling dynamic analysis, this approach overcomes the key constraints of traditional Raman instrumentation, representing a significant advancement for applications in tissue engineering and regenerative medicine. This technique holds promise for future non-invasive, real-time investigation of cartilage development and regeneration in both research and clinical settings. Another limitation of this study is the relatively small number of cartilage samples analyzed, which may constrain the robustness and generalizability of the findings. For more comprehensive scientific evaluations, future investigations should incorporate a larger and more diverse set of cartilage specimens to validate the observed trends and to enhance the statistical power. Due to the limited access to healthy material of elderly individuals, the location of the macroscopic intact and degenerated tissue is different. The location differences could contribute to the identified differences in GAG ratios. However, the disease-related pathological changes outweigh the differences in the ECM derived from the different locations. The Raman spectroscopic analysis of sGAGs is, however, limited to non-calcified tissue. In calcified tissues the sGAG peak is partially overlapped by the B-type carbonate peak and does not allow differentiation between sGAGs and B-type carbonate [15].

## 4. Materials and Methods

### 4.1. Articular Cartilage Tissue Harvest

Full-thickness cartilage comprising the superficial, middle, and deep zones of articular cartilage tissue from the tibial plateau of young individuals at different stages of skeletal maturity (immature, 4 years old, and mature young adult, 21 years old, <9 days post-mortem) was kindly provided by the Clinic for Forensic Medicine. To compare these samples to elderly and diseased tissue, full-thickness articular cartilage tissue was also harvested from the macroscopically intact femoral condyle (59 years old and 77 years old) and from the macroscopically degenerated area of the tibial plateau (59 years old) of patients undergoing knee replacement surgery at the Department of Orthopedic Surgery after obtaining informed written consent. All tissues were stored at 4 °C prior to sample harvest. To remove proteoglycans and sGAGs from the macroscopically intact cartilage (59-year-old OA patient), the tissue was digested with pronase (1 mg/mL, 7 U/mg, Roche, Mannheim, Germany) at 37 °C overnight. The samples without enzymatic digestion were incubated in DMEM high glucose containing antibiotics at 37 °C overnight.

### 4.2. Histological Stainings

For histological processing, cartilage samples were fixed (4% formalin, Otto Fischar GmbH & Co. KG, Saarbrücken, Germany), dehydrated, and paraffin embedded following the standard procedure. Thin sections were cut (Hyrax M55 Microtome, Zeiss, Oberkochen, Germany), mounted on glass slides (Superfrost^®^), and dried overnight at 60 °C. Before proceeding with histological stainings, sections were deparaffinized and rehydrated according to the standard protocol.

To visualize sGAGs, sections were stained with Safranin-O/fast green resulting in a red staining of all sGAGs. Briefly, sections were stained with fast green (0.1% in 1.2% picric acid, 1 min, Waldeck, Münster, Germany), washed with acetic acid (1.0%, 30 sec, Carl Roth, Karlsruhe, Germany), stained with Safranin-O (2.0% in distilled water, 30 min, Applichem GmbH, Darmstadt, Germany), and dehydrated.

To discriminate between strong sulfated sGAGs and non-sulfated GAGs, Alcian PAS staining was performed at pH 1.0. Sections were rinsed in hydrochloric acid (0.1 N HCl, 3 min, Carl Roth, Karlsruhe, Germany) or acetic acid for pH 2.5 (3.0% acetic acid, 3 min, Carl Roth, Karlsruhe, Germany), stained with Alcian blue (1.0% in 0.1 N HCl pH 1.0, Carl Roth, Karlsruhe, Germany), and washed with the respective solvent. Slides were pressed onto filter paper and stained with periodic acid (0.5%, 5 min, Carl Roth, Karlsruhe, Germany). After washing with tap water and distilled water, sections were incubated with Schiff’s reagent (20 min, Carl Roth, Karlsruhe, Germany), rinsed with sulfite water (0.5% sodium disulfide in distilled water containing 1N HCl, three changes à 2 min), washed with running tap water (10 min), and dehydrated.

Hematoxylin and eosin (H&E) staining was performed to visualize the tissue structure and cell nuclei. Slides were stained with Hemalum solution acid according to Mayer (8 min, Carl Roth, Karlsruhe, Germany), washed with running tap water (10 min), counterstained with Eosin Y (2 min, 0.5% in water, Carl Roth, Karlsruhe, Germany), and dehydrated. Dehydrated slides were cover-slipped using Kanadabalsam (Carl Roth, Karlsruhe, Germany) and then imaged using the Axiovert microscope (Zeiss, Oberkochen, Germany).

### 4.3. Raman Spectroscopic Line Scans

Full-thickness articular cartilage samples were fixed in ethanol (70%, 4 °C, Carl Roth Karlsruhe, Germany). A cross-section (0.5–1.5 mm) of the tissue was transferred between two slides, fixed in a slide press, and dried at 60 °C overnight. The Raman spectra were obtained with a confocal Raman spectrometer (Senterra Bruker Optik GmbH, Ettlingen, Germany) from the cross-section of the cartilage sample starting from the superficial layer towards the deep zone. A continuous laser beam (excitation: 785 nm, laser power: 100 mW, integration time: 5 s, co-additions: 10) was focused on the sample through a microscope (Olympus BX51, objective 20×, NA 0.4) and a line scan was used to measure the full thickness of the cartilage sample. To address orientation effects, we employed a depolarizer in all Raman measurements to minimize polarization-dependent intensity variations. The laser probed an area of 1 × 1 µm (measurement area), and Raman spectra were acquired every 30 µm along the line scan (step size). All Raman spectra were obtained in confocal mode (1 μm below the biopsy surface, FlexFocus, Bruker Optics, Ettlingen, Germany). To ensure reliable wavelength stability, the Senterra Raman spectrometer utilizes the Sure_Cal technology. Each Raman spectrum includes simultaneous recording of the laser excitation line and neon lamp emission lines, enabling automatic detection and correction of spectrograph and laser drift. This continuous, self-calibration process eliminates the need for external standards or manual recalibration, ensuring consistent spectral accuracy throughout the instrument’s operation.

First, we performed three line scans at three locations within the macroscopic intact cartilage from posterior condyles of the immature (4-year-old), mature young adult (21-year-old) and mature (77-year-old) donor to evaluate the reproducibility of line scans across the tissue section. In the second part, we compared the line scans of cartilage tissue harvested from one OA patient (59 years) from two different locations (macroscopic intact femoral condyle and macroscopic degenerated tibial plateau) to investigate the disease-associated local changes throughout the cartilage cross-section. The macroscopic intact cartilage was measured native and after pronase digestion to evaluate the specificity of the sGAG band.

### 4.4. Raman Spectroscopic Data Analysis

Data analysis was performed with the Opus Ident software package (OPUS 6.5, Bruker Optik GmbH, Germany). The Raman spectra were cut (350–1800 cm^−1^) and baseline corrected (rubber band, five iterations) to account for fluorescence. For quantitative evaluation of the Raman spectra, the integrated areas of the sGAGs, total GAGs (sulfated and non-sulfated GAGs), and the organic matrix (CH_2_ band) were calculated for each Raman spectrum at every point of the line scan [15]. The sGAG content was calculated from the peak at ~1062 cm^−1^ (integration from 1050 to 1080 cm^−1^), a band indicative of the symmetric stretching vibration of the OSO_3_ group. The total GAG content was calculated from the band at 1372 cm^−1^ (integration from 1370 to 1380 cm^−1^; unbranched chains of repeating sugar molecules, polysaccharides, and the CH_3_ symmetric deformation). The content of sGAG and total GAGs were normalized to the CH_2_ (integration from 1430 to 1480 cm^−1^) band [25,26]. The resulting ratios of sGAG/CH_2_ and total GAG/CH_2_ were plotted over the relative distance (percentage to total cartilage thickness). The expression of the scanned distance as a percentage of the total cartilage thickness (0–100%) was chosen to compensate for donor-to-donor variation and location-dependent differences in the total thickness of the articular cartilage samples. Representative Raman spectra (350–1800 cm^−1^) were normalized to the respective CH_2_ band (Figure 1 and Figure 3). No normalization or additional spectral manipulation (such as smoothing) was performed during the calculation of Raman spectra.

### 4.5. Statistical Analysis

For statistical analysis (GraphPad Prism Version 10.4.2), Raman ratios (sGAG/CH_2_ and total GAG/CH_2_) were plotted along the normalized tissue depth. Where appropriate, a linear regression model was applied. In cases where a linear fit was not suitable, a non-linear one-phase decay model was used to capture the distribution trend. Statistical significance was determined by calculating the corresponding *p*-values comparing the slope of the linear regression line between line scans of all donors. A threshold of *p* < 0.05 was considered statistically significant.

## 5. Conclusions

Raman spectroscopy offers a sensitive methodology for distinguishing between sGAGs and total GAGs in cartilage tissue. The high sensitivity and spatial resolution enable the characterization of local changes in the non-collagenous component of articular cartilage.

## Figures and Tables

**Figure 2 ijms-26-09875-f002:**
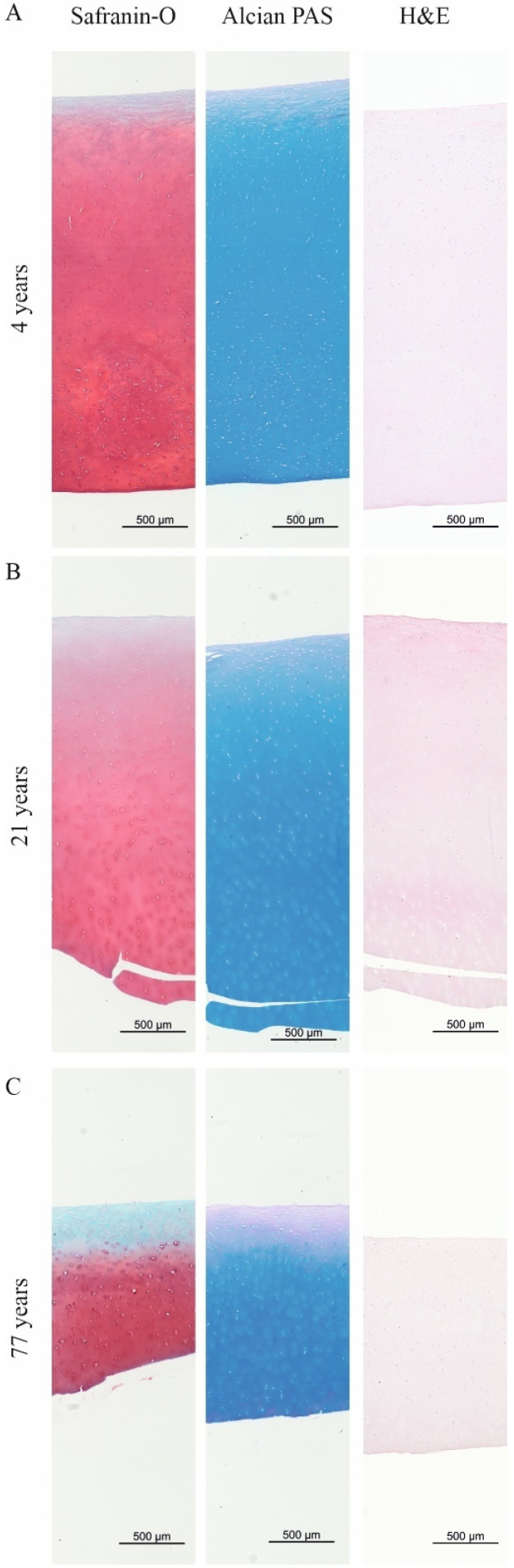
Histological stainings to visualize sGAGs and total GAGs in (**A**) skeletally immature (4 years), (**B**) skeletally mature young adult (21 years), and (**C**) skeletally mature elderly adult (77 years) articular cartilage. Safranin-O staining visualizes sGAGs (stained in red), showing a homogenous distribution throughout the cross-section in the immature (4 years) and mature (21 years) samples. The mature degenerated cartilage showed a loss of sGAG from the superficial layer to the middle zone. Alcian PAS staining to visualize sGAGs (pH 1.0) to differentiate between sulfated and strong sGAGs. Alcian PAS staining showed a similar trend as Safranin-O stained sections. Strong sGAGs (blue stain) are located in the lacunae of chondrocytes in intact cartilage. Neutral stained GAGs (purple stain) are present in the superficial zone in all samples. The strong sGAGs are found in the extracellular matrix throughout the cartilage layers. The tissue morphology was comparable between the groups. Scale bar: 500 µm.

## Data Availability

The raw data underlying the presented data in this study will be made available upon request.

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
