# Peer review of "Spatial Raman Spectroscopy to Characterize (Sulfated) Glycosaminoglycans in Human Articular Cartilage"

_ijms, 2025, doi:10.3390/ijms26209875_

Round 1

Reviewer 1 Report

Comments and Suggestions for Authors

Major Comments

  1. The study has only one donor for each group (4 y, 21 y, 77 y, and one OA donor at 59 y). Many line scans are taken from each donor, but these points are not independent. This can make very small p-values (p < 0.0001) less reliable. Authors should: Report number of sections per donor, number of lines per section, and number of spectra per line. Treat depth profiles as repeated measures (points inside line, lines inside section, sections inside donor). Use mixed-effects models with random intercepts/slopes. Show effect sizes and confidence intervals. Since only one donor is in each group, describe these as case studies and avoid strong claims.
  2. Authors assign 1370-1380 cm-1 band to only “total GAG,” but it may also include collagen or aliphatic signals. Please provide control data or references, show difference spectra, and discuss possible overlap.
  3. Authors normalize to the CH₂ band (1430–1480 cm⁻¹). But CH₂ can change with collagen density or orientation. Complementary normalizations (e.g., amide I, total area) should be tested to show that conclusions stay the same.
  4. Authors should report bounds, baselines, and integration windows. Examples of raw and processed spectra for each zone are also needed.
  5. Authors state that 30 µm step size gives 1 µm resolution. Optical resolution may be ~1 µm, but step size is not the same. This needs correction. Authors should also report axial resolution, confocal pinhole, and objective NA.
  6. Authors should report laser power at the sample, not only “100 mW nominal,” and confirm there are no heating effects. Details on cosmic-ray removal, wavenumber calibration (e.g., Si 520.7 cm⁻¹), spectral resolution, and confocal settings should be given.
  7. Authors used rubber-band baseline, five iterations, no smoothing. The reason for these choices should be explained. A flowchart or OPUS macro settings would help others repeat the method. Authors should also say if fluorescence changed with depth.
  8. Authors used pronase on cartilage. Pronase is a broad protease, not GAG-specific. Loss may be from proteoglycan release, not GAG break. Authors should explain this choice and, if possible, use chondroitinase ABC or at least discuss this limit with references. Quantitative histology matched to Raman would also help.
  9. Authors say non-linear total-GAG/CH₂ profiles are signs of immature/mature tissue and missing in OA/aged. But since the 1370–1380 cm⁻¹ band may overlap and normalization may change, this claim should be more careful. Support with histochemistry (Safranin-O/Alcian) or FTIR maps would be stronger.
  10. Authors fixed samples in ethanol and dried them. This can change Raman signals. Authors should discuss these effects, or show data comparing with fresh-frozen tissue, or cite studies showing no bias. Information on post-mortem time and storage is also needed.
  11. Authors need to fix all “Error! Reference source not found.” Cross-references. Axis labels, units, and equations with CIs should be included. Figures should have scale bars, zone marks, and line-scan paths.

Minor Comments

  • Use “one-phase exponential” and give the equation (y = A + B·e^(−x/τ)) instead of “one-phase decay.”
  • Write units consistently: cm⁻¹ (not cm-1), with space between number and unit (5 s, 100 mW, 1 µm).
  • Define how superficial/middle/deep zones are chosen (e.g., % thickness).
  • Report objective NA.
  • Confirm laser exposure is below safety limits.
  • Claims about sGAG zonality and sulfation should be supported with more references and data.
  • Words like “Reveals” or “Across Age and Disease” are too strong. With one donor per group, “Case-series evidence…” is better.
Comments on the Quality of English Language

Please correct the grammar and wording mistakes (for example: “organized hierarchically structured,” “the the,” “on phase decay,” “miocroscope,” “pronase digestion at 37°C over-night”). A full language check is needed to make the writing clear and professional.

Author Response

Thank your for your feedback and comments on our manuscript. 

In case the formatting of the response is missing the figures and, please find the point-by-point response in the attached file.

Major Comments

The study has only one donor for each group (4 y, 21 y, 77 y, and one OA donor at 59 y). Many line scans are taken from each donor, but these points are not independent. This can make very small p-values (p < 0.0001) less reliable. Authors should: Report number of sections per donor, number of lines per section, and number of spectra per line. Treat depth profiles as repeated measures (points inside line, lines inside section, sections inside donor). Use mixed-effects models with random intercepts/slopes. Show effect sizes and confidence intervals. Since only one donor is in each group, describe these as case studies and avoid strong claims.

We thank the reviewer for the comment and suggestion. The number of line scans is now reported in the figure legends and the data of all line scan is plotted in the respective figures (Figure 1 and 3). This means that each regression line corresponds to one line scan.

Figure 1: “…Number of measurement points for line-scans: 4 years (72/71/78), 21 years (83 /76/61), 77 years (42/47/39).”

Figure 3: “…Number of measurement points for line-scans: intact (104), degenerated (64), pronase treated (71).”

The main aim of the manuscript is to demonstrate the feasibility and potential of the tissue depth dependent sGAG and total GAG distribution in non-calcified cartilage at different skeletal maturity and age. Our data lays the groundwork for future studies with lager patient cohorts for a more in depth characterization of tissues across age and disease. To not overrate our findings we changed the manuscript title to “Spatial Raman Spectroscopy to characterize (Sulfated) Glycosaminoglycans in Human Articular Cartilage” and we removed strong claims in the conclusion of the manuscript.

We agree with the reviewer that the points for each line scan are dependent. However, the reported p-values results from the comparison of the slope of the lline scans of the three donors in the respective graphs (independent samples). For more clarity, we added this information in the statistical analysis section “”and in the legend of Figure 1 and 3:“The slope of the linear regression was different comparing between samples (p<0.0001)”

The three line sans per sample in Figure 1 were chosen to compare the biological variability in one sample. For the comparison of the sGAG/CH2 and total GAG/CH2 ratios, the mean ratio of the three line scans was used. Given the limited sample size, we decided to use linear regression for statistical analysis, avoiding the complex statistical models that may not be robust under these conditions.

Authors assign 1370-1380 cm-1 band to only “total GAG,” but it may also include collagen or aliphatic signals. Please provide control data or references, show difference spectra, and discuss possible overlap.

Thank you for your comment on the possible overlap of the GAG band with other signals. In Gamsjaeger et al. (2014) Journal of Raman spectroscopy http://onlinelibrary.wiley.com/doi/10.1002/jrs.4552/abstract, we investigated the 1370 cm-1 Raman band in more detail and we compared reference spectra from collagen reference and various proteoglycans. These data provide the basis for the interpretation of the spectral features discussed in this manuscript. Moreover, in Gamjaeger et al., Bone (2014) https://pubmed.ncbi.nlm.nih.gov/25245203/, we established pediatric reference Raman data for iliac trabecular bone, including graphs with references for fatty tissue, collagen, and bigylcan, which are relevant for contextualizing the spectra from patients analyzed here. We hope this clarifies the provenance of the spectral data and the rationale behind our presentation choices. This data shows that there is no overlap of the 1370 cm-1 band with the collagen band.

Gamsjaeger et al. Journal of Raman spectroscopy (2014)

Gamjaeger et al., Bone (2014)

Authors normalize to the CH₂ band (1430–1480 cm⁻¹). But CH₂ can change with collagen density or orientation. Complementary normalizations (e.g., amide I, total area) should be tested to show that conclusions stay the same.

We appreciated the reviewer’s comment regarding the orientation dependent Raman bands, particularly the amide I vibration. In our study, we addressed this issue by incorporating a depolarizer into our experimental setup. This allows us to minimize the influence of molecular orientation on Raman scattering intensity, thereby enabling a more isotropic measurement.

The amide I band is well known for its strong orientation dependence due to its vibrational mode involving C=O stretching. To further investigate this effect, we calculated intensity ratios involving the amide I band from a turkey leg tendon published in Gamsjaeger et al. Bone (2010) http://www.ncbi.nlm.nih.gov/pubmed/20450992. The data showed that the amide I Raman band is orientation dependend.

In a study by van Gulick et al. Scientific Reports (2019) https://doi.org/10.1038/s41598-019-43636-2 it is reported that the 1453 and 1270 cm−1 bands do not show any preferential orientation while amide I, amide III as well as the bands related to proline and hydroxyproline are highly sensitive to polarization.

To proof that the CH2 band in our study is not dependent on the collagen orientation, we plotted the intensity ratio of the sulfated GAGs and total GAGs normalized to the amide I band. Comparing the sGAG/CH2 and total GAG/CH2 ratios to the respective graphs of the sGAGs/amide I and total GAG/amide I, we found a similar behavior. To clarify that the CH2 band is not affected by the collagen orientation we added the following sentence in the methodology and include that for our measurements we have used a depolarizer:

Page 12: “To address orientation effects, we employed a depolarizer in all Raman measurements to minimize polarization-dependent intensity variations.”

Authors should report bounds, baselines, and integration windows. Examples of raw and processed spectra for each zone are also needed.

Thank you for your suggestion to include the missing information on spectral processing in the revised manuscript. The Raman spectra were cut (350-1800 cm-1) and afterward base lined.

Page 12: “The sGAG content was calculated from the peak at ~1062 cm-1 (integration from 1050-1080 cm-1), a band indicative of the symmetric stretching vibration of the OSO3 cm-1 group. The total GAG content was calculated from the band at 1372 cm-1 (integration from 1370-1380 cm-1; unbranched chains of repeating sugar molecules polysaccharides the CH3 symmetric deformation). The content of sGAG and total GAGs were normalized to the CH2 (integration from 1430 to 1480 cm-1) band [25, 26].“

Thank you for your comments regarding the presentation of the raw Raman spectra. In the graph below, we plotted all measurements from a single line scan of the 4 year old patient. This graph is very busy and small differences in the sGAG and total GAG peaks are not visible. Therefore, we decided not to show all raw single spectra in the main manuscript. Since all spectra show a similar trend, we selected one representative graph for the superficial, middle and deep zone shown in Figure 1 and 3. Raw data is available upon request.

Authors state that 30 µm step size gives 1 µm resolution. Optical resolution may be ~1 µm, but step size is not the same. This needs correction. Authors should also report axial resolution, confocal pinhole, and objective NA.

We thank you for this comment and clarified the term resolution and confocal acquisition settings. Throughout the manuscript, we included the term “step size” and/or “measurement area” when referring to the resolution of the Raman scans.

Page 12: “The laser propped an area of 1x1 µm (measurement area), and Raman spectra were acquired every 30 µm along the line scan (step size)…. (Olympus BX51, objective 20×, NA 0,4) …. All Raman spectra were obtained in confocal mode (1 μm below the biopsy surface, FlexFocus, Bruker Optics).”

Authors should report laser power at the sample, not only “100 mW nominal,” and confirm there are no heating effects. Details on cosmic-ray removal, wavenumber calibration (e.g., Si 520.7 cm⁻¹), spectral resolution, and confocal settings should be given.

We appreciate your concern regarding the possible heating effects. We have been using this Raman protocol since 2010. To assess reproducibility and potential effects of tissue quality and thermal impact, we performed 25 repeated measurements on the same tissue area using a laser power of 100 mW published in Gamsjaeger et al. 2014, Bone, https://pubmed.ncbi.nlm.nih.gov/25245203/). In Gamsjaeger et al. 2014, we included a detailed table reporting the technical variance (%COV) for each parameter. The low variance observed across these measurements indicated high reproducibility and no thermal effect. If heat had significantly influenced the measurements, we would expect a substantially higher %COV. We have followed the same methodological approach in the current study, ensuring consistency with previously validated protocols.

In terms of spectral integrity, we avoid automated spectral manipulations, including cosmic ray removal algorithms. If a cosmic ray artifact was observed within the spectral range of interest (350-1800 cm-1), the spectrum was discarded and reacquired. This approach ensures that all spectra included in the analysis are free from such artifacts without relying on post-processing corrections.

For wavenumber calibration and spectral stability, we have added the following description:

Page 12: “To ensure reliable wavelength stability, the Senterra Raman spectrometer utilizes the Sure_Cal technology. Each Raman spectrum includes simultaneous recording of the laser excitation line and Neon lamp emission lines, enabling automatic detection and correction of spectrograph and laser drift. This continuous, self-calibration process eliminates the need for external standards or manual recalibration, ensuring consistent spectral accuracy throughout the instrument’s operation.“

Authors used rubber-band baseline, five iterations, no smoothing. The reason for these choices should be explained. A flowchart or OPUS macro settings would help others repeat the method. Authors should also say if fluorescence changed with depth.

Thank you for your feedback. Base line correction was performed using the standard baseline correction tools available within the OPUS spectroscopy software (Bruker), without the use of custom macros. This approach was consistently applied across multiple versions of OPUS, ensuring methodological reproducibility. As OPUS is a commercially available and widely adopted platform for infrared and Raman spectroscopy, the use of its basline correction functions ensures transparency and accessibility and allows inter-laboratory reproducibility of spectral data processing.

Authors used pronase on cartilage. Pronase is a broad protease, not GAG-specific. Loss may be from proteoglycan release, not GAG break. Authors should explain this choice and, if possible, use chondroitinase ABC or at least discuss this limit with references. Quantitative histology matched to Raman would also help.

We thank you for your comment on the use of pronase to digest proteoglycans from cartilage tissue. Pronase allows to digest the protein core and thereby non-specifically degrade all GAGs, thus hydrolyzes proteins non-specifically. In contrast, the use of e.g heparinase or chondroitinase would only digest the respective GAG side chain and would require a combination of multiple enzymes to degrade all sulfated GAG chains of chondroitin sulfate, keratin sulfate and heparan sulfate. This approach is more expensive and needs a fine-tuned protocol for successful digestion and it also requires the analysis and characterization of the digested GAG fractions.

Pronase is one of the commonly used enzymes to reproducibly and successfully degrade proteoglycans.

Since we showed the removal of sulfated GAGs (no red sulfated GAG staining) from the tissue in the Safranin-O staining of the pronase treated cartilage, we achieved our aim to remove sulfated GAGs from the tissue sections. Additionally, the purple stain in the Alcian PAS staining confirmed the removal of the sulfated GAGs compared to the non-treated cartilage tissue.

Due to the low sample number and the obvious visual differences in the Alcian PAS and Safranin-O staining, in our opinion, there is no need for more quantitative analysis of the histological staining. As discussed in the manuscript, the histological stainings are in line with the overall distribution of sulfated GAGs quantified by Raman spectroscopy. The small variation in histological stainings and Raman ratios are a result of the less specificity of the stainings based on ionic interaction of the dye with the negatively charged sGAGs and the lower resolution of histological stainings compared to the high resolution of the Raman line-scans used in this study. This was discussed in the manuscript (Page 5 and 11).

Authors say non-linear total GAG/CH₂ profiles are signs of immature/mature tissue and missing in OA/aged. But since the 1370–1380 cm⁻¹ band may overlap and normalization may change, this claim should be more careful. Support with histochemistry (Safranin-O/Alcian) or FTIR maps would be stronger.

Thank you for your feedback. In this study we focused on the establishment of a fast and accessible Raman-based method with minimal sample preparation. The 1370 cm-1 Raman peak, which is central to our analysis, is well resolved and does not suffer from spectral overlap under the chosen conditions (see answer above, figures with the references). We did not perform FTIR measurements for this analysis, as FTIR typically requires embedding the tissue and preparing 4 µm thin sections (standard protocol in our laboratory). This process is more labor-intensive compared to Raman spectroscopy. Our goal was to maintain a streamlined protocol that preserves reproducibility and reduces preparation complexity regarding histological validation that would not be compatible with FTIR analysis. Instead, we compared the sulfated GAG and total GAG ratios of the Raman analysis to Safranin-O (stains negatively charged GAGs) and Alcian PAS (differentiates negatively charged and neutral GAGs) stainings (Figure 2 and 4) of samples harvested from the same location of the respective donors showing a similar trend to the depth dependent Raman ratios. Therefore, we are confident that the ratios calculated from the Raman scans are not a result of spectral overlap nor a result of the normalization.

Authors fixed samples in ethanol and dried them. This can change Raman signals. Authors should discuss these effects, or show data comparing with fresh-frozen tissue, or cite studies showing no bias. Information on post-mortem time and storage is also needed.

We thank the reviewer for your query on the sample processing and we like to clarify the rationale behind our sample preparation protocol: All samples were treated following the same protocol prior to Raman measurements. This standardized approach was chosen to ensure reproducibility across the dataset and to minimize variability introduced by differing sample conditions. Uniform treatment allows for consistent comparison of Raman spectra and ensures that observed differences are attributable to biological variation rather than preparation artifacts.

The sample drying was necessary for practical and technical reasons: First, samples needed to be stored prior to measurements, and ethanol fixation followed by drying provided a stable procedure preserving tissue integrity. Second, Raman line scans require a flat and stable surface to maintain consistent focus and signal quality. Fresh samples tend to undergo surface changes during measurement due to evaporation, which can alter the Raman signal within minutes. By drying the samples in advance under controlled conditions, we ensured a uniform surface and avoid such dynamic changes during acquisition. We are aware that Raman spectra obtained from ethanol fixed and dried tissue differ from those acquired from fresh tissue. We hope that it is now clearer why we used this protocol.

The cadaveric tissue samples were kept at 4°C until the legally conducted autopsy. After opening the knee joint, the articular cartilage was harvested and immediately fixed with 70% EtOH at 4°C for Raman spectroscopy or formalin fixed and paraffin embedded for histological stainings. Tissue from patients undergoing knee replacement surgery was stored over-night at 4°C before the tissue was harvested and fixed (70%EtOH at 4°C)

We added the information of the post-mortem time in the revised manuscript:

Page 11: “Full thickness cartilage comprising the superficial, middle and deep zone of articular cartilage tissue from the tibial plateau of young individuals at different stage of skeletal maturity (immature, 4 years old and mature young adult, 21 years old, <9 days post-mortem)… All tissues were stored at 4°C prior to sample harvest. To remove proteoglycans and sGAGs”.

Authors need to fix all “Error! Reference source not found.” Cross-references. Axis labels, units, and equations with CIs should be included. Figures should have scale bars, zone marks, and line-scan paths.

Thank you for spotting the formatting errors. We carefully went through the manuscript to correct these errors with particular attention on the citation accuracy, consistency with the reference list, and alignment with the journal’s formatting guidelines.

Minor Comments

Use “one-phase exponential” and give the equation (y = A + B·e^(−x/τ)) instead of “one-phase decay.”

Thank you for sharing your opinion. We decided to use the term one-phase decay, as this nomenclature is consistent with the terminology used in GraphPad Prism, the software we and many other gorups are using for data analysis. In Prism, one phase decay refers specifically to an exponential decrease from an initial value towards a plateau at a constant rate. This terminology is widely recognized by users of the software and accurately refers the mathematical model applied to our data. The use of this term ensures clarity and reproducibility for readers familiar with Prism and facilitates direct comparison with similar studies using the same analytical framework.

Page 13: “For statistical analysis (GraphPad Prism Version 10.4.2), Raman ratios (sGAG/CH2 and total GAG/CH2) were plotted along the normalized tissue depth.”

Write units consistently: cm⁻¹ (not cm-1), with space between number and unit (5 s, 100 mW, 1 µm).

All unit notations have been corrected in the revised manuscript.

Define how superficial/middle/deep zones are chosen (e.g., % thickness).

We thank the reviewer for raising this point. In the current study, we did not explicitly define the superficial, middle, and deep zones based on percentages, thickness or other quantitative criteria. Instead, we chose the representative spectra in each zone based on the histomorphometric characteristics of each sample individually.

Report objective NA.

We appreciate the reviewer’s observation. The necessary information regarding NA has been incorporated into the revised manuscript.

Confirm laser exposure is below safety limits.

Thank you for raising the safety concerns. The Senterra Raman microscopy is a commercially available instrument that complies with established safety regulation standards. During measurements, the enclosure surrounding the microscope remains securely closed, ensuring that no external light can enter, and no laser radiation can escape, thereby maintains a safe operating environment. The system is integrated with a quality management interface that provides automated notifications when laser replacement is required. Regular maintenance is performed in accordance with manufacturer guidelines to ensure optimal performance and continued compliance with safety and operational standards.

Claims about sGAG zonality and sulfation should be supported with more references and data.

We thank you for your suggestion and added a few more references in the respective paragraphs in the discussion. In the introduction we cited a selection of literature on the zonality of GAGs and specifically the sulfation to support the rationale. Of course, there is more knowledge on the specific sulfated GAG distribution in literature. However, this exceeds the scope of this manuscript, the use of Raman line-scans to assess the depth dependent sGAG and total GAG in non-calcified articular cartilage tissue. We carefully read through the discussion of the manuscript and added additional references where needed.

Words like “Reveals” or “Across Age and Disease” are too strong. With one donor per group, “Case-series evidence…” is better.

Thank you for your critical feedback. To not overrate our results, we revised the title of the manuscript to “Spatial Raman Spectroscopy to characterize (Sulfated) Glycosaminoglycans in Human Articular Cartilage”

The limitation of the low sample number has been acknowledged at the end of the discussion section (see also comments on this before). The findings should be interpreted as preliminary but indicative, and they provide a foundation for future studies with expanded cohorts.

Page 11: “Another limitation of this study is the relatively small number of cartilage samples analyzed, which may constrain the robustness and generalizability of the findings. For more comprehensive scientific evaluations, future investigations should incorporate a larger and more diverse set of cartilage specimens to validate the observed trends and to increase the statistical power.”

Reviewer 2 Report

Comments and Suggestions for Authors

General comments:
Scarce explanation about what a Raman spectroscopy graphic really displays (what the axis represent). I think the authors could explain better the physics behind this technique.

No explanation about how the 3 cartilage zones can be separated/distinguished using Raman spectroscopy. Did the authors just divide the cartilage length (0-100%) in 3 equal parts? If so, why? Moreover, it is not clear if they include the calcified cartilage zone in the measured sections. If not, why? Calcified cartilage zone can be stained by DMMB and Safranin-O stainings, and hence they contain sGAGs too. 

Very low numerosity (n=1-2), only 3 technical replicates per sample. This might be a limitation to claim that the authors found specific Raman spectroscopy patterns for different cartilage conditions.

H&E stainings do not really provide more information, cartilage appears almost unstained

A lot of comments about Raman line-scans duration, but no comments about costs

Figures at page 4 and 8 are both called "Figure 1"

Specific comments: 

Lines 36 and 107: no mention about calcified cartilage zone
Line 80-81: No explanation about the meaning of parameters (e.g. 1062 cm^-1) 

Page 4, Figure 1A and D, Figure; Page 8, Figure 1A, B and C: 77 yo cartilage sample sGAG/CH2 ratio and CH2 zone-specific spectra are very similar to the sGAG/CH2 ratio and CH2 zone-specific spectra of intact and degenerated OA cartilage samples. No comments about this? 

Page 8, Figure 1B: sGAG peaks from different zones have all the same values, while the same cartilage stained with Safranin-O shows a loss of sGAGs in the superficial zone, but not in the deeper ones (Page 10, Figure 4A).

Figure 2:  Please show the full thickness cartilage to the bone. Do the different cartilage samples come from the same region of the knee or from different ones? If it is not known this should be stated as the difference can also be localization-dependent.

Line 207: "fibrillation were also visible in H&E staining" where? All H&E stainings are almost white. 
Lines 216, 217, 218: I can see a difference in sGAGs between superficial and middle/deep zone, looking at Stained sections. 
Lines 255-256: It is not true that enzymatic digestion removed GAGs from cartilage tissue, they are still detectable in Raman spectroscopy (Page 8, figure 1D)
Lines 265-270: Chondroitin sulphate, keratan sulphate and heparan sulphate are sGAGs and they cannot be differentiate with Raman spectroscopy and neither with Raman spectroscopy line scan, that discriminated only between sGAGs and GAGs. 
Line 308: I don't see how having a microscope mounted on a Raman spectrometer could be a limitation. 

Author Response

Reviewer 2

General comments:

Scarce explanation about what a Raman spectroscopy graphic really displays (what the axis represent). I think the authors could explain better the physics behind this technique.

Thank you for your comment. We revised the description of the axis labelling to align with the Journal of Raman spectroscopy. Additionally, we modified the Introduction to include a brief explanation of the underlying physics of Raman spectroscopy to provide a better context for readers less familiar with the technique and strengthens the foundation for the methodological focus of our study.

Page 2: “Raman spectroscopy relies on the inelastic scattering of photons, where incident light interacts with molecular vibrations, leading to energy shifts that reveals the vibrational modes of chemical bonds.”

The y-axis in our Raman spectra does not present absolute intensity values, as we have normalized all spectra to the CH2 Raman band. This normalization was applied for a better visibility and to account for variations in signal strength, allowing for consistent comparison across samples. We followed the guidelines from the Journal of Raman spectroscopy suggesting: “Figure intensity scales for Raman spectra should be "Raman Intensity/Arbitr. Units". However, use "/Arbitr. Units" only if units/numbers are actually given on the vertical scale; otherwise use "Raman Intensity" only. Do not use “a.u.” on the intensity scale to avoid confusion with atomic units.“ Therefore, we use the term “Raman intensity” in a relative sense when reporting and discussing the data.

No explanation about how the 3 cartilage zones can be separated/distinguished using Raman spectroscopy. Did the authors just divide the cartilage length (0-100%) in 3 equal parts? If so, why? Moreover, it is not clear if they include the calcified cartilage zone in the measured sections. If not, why? Calcified cartilage zone can be stained by DMMB and Safranin-O stainings, and hence they contain sGAGs too.

We thank you for raising these questions. The full thickness articular cartilage samples used in this study comprise the superficial layer (unless degraded due to OA disease), middle zone and deep zone. In our study, we did not divide the cartilage into the three sections for the analysis. Instead, we acquired Raman scans at a step size of 30 µm throughout the full thickness cartilage tissue starting in the superficial zone towards the deep zone.

The cartilage length in the diagrams of figure 1 and 3 are expressed as % to the total thickness to compensate for the differences in cartilage thickness due to age and disease. The representative Raman spectra for the superficial layer, middle and deep zone shown in Figure 1 and 3 were selected based on the tissue structure and cell morphology. To select the representative spectra, we oriented ourselves by the following tissue characteristics: The superficial zone comprises the upper 10-20% of the cartilage thickness with flat cells oriented vertically. The middle zone (40-60% of the total thickness) represents the largest area of the non-calcified cartilage with round chondrocytes being randomly oriented in the tissue. The main characteristics of the deep zone are the columnar multicellular organization in lacunae.

Based on our data, it is not yet possible to discriminate between the different zones using the parameters we investigated.

The histological stainings clearly show that the cartilage characterized in this study comprises the three zones of the non-calcified cartilage, but not the zone of calcified cartilage. To clarify the zones of the cartilage cross-section, we added the following sentence:

Page 11: “Full thickness articular cartilage comprising the superficial, middle and deep zone of articular cartilage tissue….”

Due to the overlap of the sulfated GAG and the B- type carbonate peak in the Raman spectra, sulfated GAG measurements are limited to non-calcified tissues. To clarify that our cartilage did not contain the zone of calcified cartilage we added this information to the materials and methods section.

Page 11: “The Raman spectroscopic analysis of sGAGs is, however, limited to non-calcified tissue. In calcified tissues the sGAG peak is partially overlapped by the B-type carbonate peak and does not allow to differentiate between sGAGs and B-type carbonate [15].”

Very low numerosity (n=1-2), only 3 technical replicates per sample. This might be a limitation to claim that the authors found specific Raman spectroscopy patterns for different cartilage conditions.

Thank you for your concerns regarding the technical replicates. As noted in the last paragraph of the Discussion, we acknowledge that the relative low sample number constrain the generalizability of our findings are one limitation of the study. However, we would like to highlight that the primary aim of this study was to introduce and validate the methodological framework, rather than to draw broad statistical conclusion. The approach we present is designed to be scalable and adaptable, and we believe that even with a limited sample size, the method demonstrates its potential for future studies on a larger patient cohort.

Page 11: „Another limitation of this study is the relatively small number of cartilage samples analyzed, which may constrain the robustness and generalizability of the findings. For more comprehensive scientific evaluations, future investigations should incorporate a larger and more diverse set of cartilage specimens to validate the observed trends and to enhance the statistical power.“

H&E stainings do not really provide more information, cartilage appears almost unstained

We thank you for your opinion on the H&E staining. Indeed, the H&E stained cartilage sections are lightly colored. The reason for the light staining can be explained by the interaction of the dyes with the tissue. The basic Hematoxylin dye binds to the negative charge of the sulfated GAGs in the cartilage tissue resulting in the light blue staining, whereas the acidic Eosin can only be attracted and bind to neutral to positive charged matrix proteins. This explains the more intense purple staining in the pronase treated intact cartilage samples and in the fibrillated superficial/middle layer in the OA degenerated sample in figure 4. The reason why we decided to show the H&E stainings is to visualize the tissue morphology and the tissue boundaries (e.g fibrillation).

A lot of comments about Raman line-scans duration, but no comments about costs

Thank you for your comment on the costs for line-scans. We introduced Raman line-scans in our manuscript to be less time-intensive compared to full Raman images, which makes them a more practical choice on many experimental settings. We agree with you that a cost estimation might be also of interest for the reader. However, the costs for line scans are influenced by several external factors, including the salary structure of an institution and the specific pricing of the Raman instrument used, which can vary widely across manufacturers and regions.

Figures at page 4 and 8 are both called "Figure 1"

Many thanks for pointing this out. We have corrected the Figure numbers.

Specific comments: 

Lines 36 and 107: no mention about calcified cartilage zone

Thank you for your comment to clarify on the composition of the full thickness articular cartilage samples. As discussed earlier the “Full thickness cartilage comprising the superficial, middle and deep zone of articular cartilage tissue” since the focus of this study was to characterize the sGAG and GAG distribution in non-calcified cartilage tissue. As shown in our publication (Gamsjaeger et al. 2014, https://analyticalsciencejournals.onlinelibrary.wiley.com/doi/full/10.1002/jrs.4552), the Raman band at 1060 cm-1, commonly used to assess GAG content, is overlapped by the B-type carbonate signal in calcified cartilage. This spectral interference makes reliable sGAG quantification in calcified regions problematic. For this reason, we chose not to investigate calcified cartilage in this study to ensure accurate and interpretable Raman measurements.

Page 11: “The Raman spectroscopic analysis of sGAGs is, however, limited to non-calcified tissue. In calcified tissues the sGAG peak is partially overlapped by the B-type carbonate peak and does not allow to differentiate between sGAGs and B-type carbonate [15].”

Line 80-81: No explanation about the meaning of parameters (e.g. 1062 cm-1

Thank you for your comment. To further improve the meaning of the Raman peaks investigated in this study we modified the text:

Page 2: “Raman spectroscopic imaging has been used to detect sGAGs from the peak at 1062 cm-1 (symmetric stretching vibration of the OSO3 cm-1 group), total GAGs from the band at 1370-1380 cm-1 (unbranched chains of repeating sugar molecules polysaccharides the CH3 symmetric deformation), and organic matrix components (CH2 band: 1430-1480 cm-1) for the distinction of tissue types, and to characterize GAG loss in articular cartilage in OA patients [15, 22, 25-27].“

Page 12: “The sGAG content was calculated from the peak at ~1062 cm-1 (integration from 1050-1080 cm-1), a band indicative of the symmetric stretching vibration of the OSO3 cm-1 group. The total GAG content was calculated from the band at 1372 cm-1 (integration from 1370-1380 cm-1; unbranched chains of repeating sugar molecules polysaccharides the CH3 symmetric deformation). The content of sGAG and total GAGs were normalized to the CH2 (integration from 1430–1480 cm-1) band [25, 26].”

Page 4, Figure 1A and D, Figure; Page 8, Figure 1A, B and C: 77 yo cartilage sample sGAG/CH2 ratio and CH2 zone-specific spectra are very similar to the sGAG/CH2 ratio and CH2 zone-specific spectra of intact and degenerated OA cartilage samples. No comments about this? 

Thank you for pointing out the missing comparison between the figures. We believe you are referring to figure 1 A&D and figure 3 A&B. Indeed, the intact sample in figure 3 A and the tissue sample from the aged patient in figure 1 D are harvested from the same location and show a similar Raman spectrum in the three zones. Due to the progressed OA disease in the 77 year old patient characterized in Figure 1 and 2, the sGAG/CH2 ratios are more similar to the degenerated cartilage tissue in Figure 3&4. This assumption is supported by the histological stainings showing the loss of sGAGs in the superficial down to the middle zone in the two donors. We did not compare the spectra between Figure 1 and 3 due to the low sample number and to not oversell our findings in this methodology based manuscript.

Page 8, Figure 1B: sGAG peaks from different zones have all the same values, while the same cartilage stained with Safranin-O shows a loss of sGAGs in the superficial zone, but not in the deeper ones (Page 10, Figure 4A).

Thank you for raising this comment. We assume you are referring to the Raman spectra of the intact cartilage tissue in figure 3B. You are right that the peak height is of the representative sulfated GAG peak is similar in the three zones. While peak intensity is often used as a proxy for concentration, it does not fully capture the spectral complexity. Therefore, we suggest that interpretation of Raman peak height for small changes should be approached with caution. To overcome this in the analysis of our data, we have considered the height and width of the Raman band, as the integrated area provides a more accurate representation of the spectral feature. For the peak quantification, we have focused on the ratio of specific bands, which we believe offers a more robust and comparative metric across samples compared to “absolute” values of a single peak.

Figure 2:  Please show the full thickness cartilage to the bone. Do the different cartilage samples come from the same region of the knee or from different ones? If it is not known this should be stated as the difference can also be localization-dependent.

Thank you for your question. Our study focuses on the sGAG and total GAG distribution in non-calcified cartilage. Therefore, we harvested the full thickness cartilage tissue (comprising the superficial, middle and deep zone) without the technically hard to cut zone of calcified cartilage and subchondral bone. As discussed before, the Raman peak for sulfated GAGs and the B-type carbonate peak are overlapping, thus are not recommended to perform Raman analysis of sulfated GAGs in the zone of calcified cartilage, nor in the bone. The presence of calcified tissue would require at least a mild decalcification step in order to cut planar tissue sections needed to analyze the tissue. Decalcification protocols, even if a mild treatment is used, has an impact on the tissue structure and composition and, therefore, we did cut off the calcified tissue.

The cartilage was taken from either the tibial plateau or the femoral condyles as described in the materials and methods section. The tissue harvest was done from the same location within the knee joint to harvest tissue with matched locations between donors. We clarified this in the revised manuscript.

Page 11: “Full thickness cartilage comprising the superficial, middle and deep zone of articular cartilage tissue from the tibial plateau of young individuals at different stage of skeletal maturity … full thickness articular cartilage tissue was also harvested from the macroscopically intact femoral condyle (59 years and 77 years) and from the macroscopically degenerated area of the tibial plateau...patients undergoing knee replacement surgery”.

We agree with the reviewer, that the differences in the sulfated GAG and total GAG ratios could partially result from the different locations the intact and macroscopically degraded cartilage was harvested. The location dependent changes in the ECM composition are minor compared to the obvious pathologic changes due to OA disease. To account for this, we added a comment on the location differences to the limitations in the discussion of the revised manuscript.

Page 11: “Due to the limited access to healthy material of aged individuals, the location of the macroscopic intact and degenerated tissue was different. The location differences could contribute to the identified differences in GAG ratios. However, the disease related pathological changes outweigh the differences in the ECM derived from the different locations.”

Line 207: "fibrillation were also visible in H&E staining" where? All H&E stainings are almost white. 

We apologize for the low contrast in the images of the H&E stained sections, a result of our standard staining protocol. We already explained the reason for the light staining in the comment before and we do not fully agree with you that the staining is almost white. All tissue sections are lightly stained and the tissue morphology is still visible.

The fibrillation in the OA cartilage tissue in Figure 4 is most obvious in panel B of the degenerated tissue. This fibrillation can be also seen in the Safranin-O and Alcian PAS stained sections, and is not limited to the H&E staining. This information is now added to the manuscript.

Page 7: “First signs of superficial fibrillation and sGAG loss were visible in the histological stainings.”

Lines 216, 217, 218: I can see a difference in sGAGs between superficial and middle/deep zone, looking at Stained sections. 

Thank you for your comment. We believe you are referring to the minimal differences in the Safranin-O &/or Alcian PAS staining in Figure 4. The shown images are from a mosaic image acquisition to increase the resolution of the imaged tissue cross section. To create these images, 3-5 images were acquired and stitched together. This can induce small differences in the color and therefore, we reported that there were no obvious differences between the zones.

Lines 255-256: It is not true that enzymatic digestion removed GAGs from cartilage tissue, they are still detectable in Raman spectroscopy (Page 8, figure 1D)

We acknowledge the presence of a Raman band at 1060 cm-1 in the pronase treated spectra, and we agree that its persistence is noteworthy. This residual signal may be attributed to the inherent sensitivity of Raman spectroscopy, the efficiency of GAG removal via pronase treatment is not absolute and depends on several key factors, including the concentration of the pronase used and the duration of incubation. Interestingly, the sGAGs are completely removed on the histological stained sections, residues of sGAG and GAG peaks are still detectable in the Raman spectra we added to the discussion

Page 9/10: „Despite enzymatic treatment with pronase, a Raman band for sGAGs and total GAGs remains detectable in the spectra. The low Raman signal suggests that the tissue contains residues of sGAGs not being detectable on histological sections. This observation highlights the high sensitivity of Raman spectroscopy over histological staining by detecting residual molecular features after biochemical tissue modification“.

Lines 265-270: Chondroitin sulphate, keratan sulphate and heparan sulphate are sGAGs and they cannot be differentiate with Raman spectroscopy and neither with Raman spectroscopy line scan, that discriminated only between sGAGs and GAGs. 

We agree with the reviewer that Raman line sans alone cannot provide sufficient discrimination of the sGAG groups. In our manuscript, we us “Raman spectroscopy” as a general term encompassing all modalities of the technique, including single point spectra, line scans, and imaging, rather than referring to any specific approach. To differentiate between the different GAGs, biochemical analysis (e.g mass spectroscopy) at high sensitivity is needed to do so. However, this aspect is out of scope for our manuscript.

Line 308: I don't see how having a microscope mounted on a Raman spectrometer could be a limitation. 

Thank you for your feedback. Microscopy-enhanced Raman spectroscopy may offer precision, but it comes with trade offs. In general, such an instrument presents limitations, particularly regarding in vivo applicability. What we intended to convey is that in vivo measurements are not feasible with the setup used in our study due to its design constraints. To clarity this point, we have revised the manuscript to explicitly include the term “in vivo” in the discussion of limitations. Furthermore, we referenced the Raman setup used by Berthold et al., which demonstrated a configuration that overcomes this specific limitation. Their approach opens the possibility for acquiring Raman spectra in a clinical context, such as on fresh tissue during surgical procedure (a future application).

Page 10: “One limitation of Raman spectroscopy is that standard spectrometers (e.g, Senterra I, Bruker used in this study) are typically equipped with a microscope, require sample preparation, and are not suitable for direct analysis of live tissue in vivo.”

Round 2

Reviewer 2 Report

Comments and Suggestions for Authors

The authors responded appropriately to all my comments. They made several important changes in their revised version that increased the clarity and improved the quality of the manuscript.